# The social media diet: A scoping review to investigate the association between social media, body image and eating disorders amongst young people

**Alexandra Dane**, **Komal Bhatia** *

Institute for Global Health, University College London, London, United Kingdom

* komal.bhatia.14@ucl.ac.uk

## Abstract

### Background

Eating disorders are a group of heterogenous, disabling and deadly psychiatric illnesses with a plethora of associated health consequences. Exploratory research suggests that social media usage may be triggering body image concerns and heightening eating disorder pathology amongst young people, but the topic is under-researched as a global public health issue.

### Aim

To systematically map out and critically review the existing global literature on the relationship between social media usage, body image and eating disorders in young people aged 10–24 years.

### Methods

A systematic search of MEDLINE, PyscINFO and Web of Science for research on social media use and body image concerns / disordered eating outcomes published between January 2016 and July 2021. Results on exposures (social media usage), outcomes (body image, eating disorders, disordered eating), mediators and moderators were synthesised using an integrated theoretical framework of the influence of internet use on body image concerns and eating pathology.

### Results

Evidence from 50 studies in 17 countries indicates that social media usage leads to body image concerns, eating disorders/disordered eating and poor mental health via the mediating pathways of social comparison, thin / fit ideal internalisation, and self-objectification. Specific exposures (social media trends, pro-eating disorder content, appearance focused platforms and investment in photos) and moderators (high BMI, female gender, and pre-existing body image concerns) strengthen the relationship, while other moderators (high

**Data Availability Statement:** All relevant data are within the paper and its Supporting Information files.

**Funding:** The authors received no specific funding for this work.

**Competing interests:** The authors have declared that no competing interests exist.

social media literacy and body appreciation) are protective, hinting at a 'self-perpetuating cycle of risk'.

## Conclusion

Social media usage is a plausible risk factor for the development of eating disorders. Research from Asia suggests that the association is not unique to traditionally western cultures. Based on scale of social media usage amongst young people, this issue is worthy of attention as an emerging global public health issue.

## Introduction

### Eating disorders in the 21st century

**Types and burden.** Eating disorders are a group of heterogenous, disabling and deadly psychiatric illnesses that severely impair daily psychological and social functioning [1]. Characterised by disturbed body image attitudes and extreme preoccupations with weight and shape, eating disorders manifest as persistent and worrisome disordered eating behaviours [2]. International ICD-11 and DSM-5 diagnostic classification tools recognise six principal clinical eating disorders [**Table 1**] [3]. A supplementary Other Specified Feeding and Eating Disorder

**Table 1. Types of clinical and subclinical eating disorders and typical pathology.**

| CLINICAL EATING DISORDERS | |
|---|---|
| **Anorexia** | An intense fear of weight gain and/or a disturbed body image that motivates severe dietary restriction or other weight loss behaviours |
| **Bulimia** | Recurrent episodes of binge eating and compensatory behaviours, e.g., purging, to prevent weight gain |
| **Binge eating disorder** | Recurrent episodes of compulsive overeating that leads to distress without attempts to compensate for weight gain |
| **Avoidant/restrictive food intake disorder** | The avoidance or restrictive intake of food in the absence of body image concerns and fear of weight gain |
| **Pica** | Eating non-nutritive or non-food substances for a period of one month or more |
| **Rumination disorder** | Involves regurgitation of food after eating in the absence of nausea, involuntary retching, or disgust |
| **SUBCLINICAL OTHER SPECIFIC FEEDING AND EATING DISORDERS** | |
| **Orthorexia Nervosa** | A pathological fixation with healthy or 'clean' eating, avoidance of unhealthy foods and rigid dietary and exercise practices- violations of which cause severe emotional distress |
| **Atypical anorexia** | Majority of symptoms of anorexia are present, but the individual is classified as being within the normal BMI range |
| **Atypical bulimia** | Mimics clinical bulimia but occurs less frequently and with shorter duration |
| **Atypical binge eating disorder** | Mimics clinical binge eating disorder but occurs less frequently and with shorter duration |
| **Purging disorder** | Purging or using laxatives as a mean to control weight |
| **Night eating disorder** | Repeatedly eating at night, either after an evening meal or waking up from sleep |
| COMMON PATHOLOGY | |
| **Dieting, binging, purging, restricting, avoidance of certain food groups, compulsive or compensatory exercise behaviours and the use of laxatives or weight loss pills** | |

Adapted from [1, 2, 5, 6]

(OSFED) category captures approximately 60% of cases that do not meet criteria for clinical diagnosis [4].

Eating disorders incur an estimated 6–10% increase in years lived with disability [7]. Outcomes range from cardiovascular disease, reduced bone density, to comorbid psychiatric conditions, namely depression, anxiety, obsessive compulsive disorder and specific phobias [8, 9]. Amongst young females, eating disorders are one of the leading causes of disability, often preceding amenorrhea, reduced fertility, and adverse pregnancy and neonatal outcomes [10, 11]. Anorexia has the highest mortality amongst all mental disorders: only 50% of individuals fully recover [7, 8, 12].

The cost of eating disorders at a health systems level is significant, fuelled by increased hospitalisations and the significant burden placed on primary and outpatient services. At a societal level, reduced workforce participation, family members as unpaid carers and young people out of education are noteworthy outcomes of eating disorders [11].

**Epidemiology.** Despite perceptions of eating disorders as a culturally bound syndrome of the West, they affect individuals worldwide [7]. Estimating global prevalence, however, is challenging. Nationally representative data are scarce, the disorder tends to be omitted from national health surveys, and multiple changes to classification have confounded existing global data [13–15]. Despite this, the most recent Global Burden of Disease study calculated that in 2019, approximately 13.9 million people suffered from Anorexia or Bulimia. A subsequent review highlighted an additional 41.9 million overlooked cases of OSFED and binge eating disorder, indicating a total global prevalence of 0.7% [2]. However, since many cases never present at formal health services, actual prevalence may be much greater [16]. A review of 94 studies from Asia, Europe and North America revealed that the weighted mean of lifetime prevalence of any eating disorder was 8.4% for women and 2.2% for men [17].

Whilst females still represent the largest proportion of cases, the greatest increase is amongst males, athletes, those with obesity, and sexual and gender minorities [18–21]. Most eating disorders begin in adolescence but tend to persist throughout adulthood [22]. Therefore, young people constitute a subgroup of particular concern [13, 17].

**Aetiology and risk factors.** The aetiology of eating disorders is complex; no single risk factor accounts for their manifestation [23]. Rather, prevalence is hypothesised to be the result of numerous biological, psychological, psychosocial, and behavioural factors [**Table 2**].

Body image, a multidimensional psychological construct encompassing how we think, feel and act towards our bodies—has been recognised as the most salient and consistent predictor of eating disorder symptomatology [9, 25, 26]. Although grounded in physical appearance, body image is rarely synonymous with it—individuals often view themselves through a lens of dysmorphia, seeing fatness, ugliness, or an endless list of flaws. The need to 'fix' what is 'faulty' is thought to precede compensatory disordered eating and appearance altering behaviours [27]. Owing to pubertal weight gain, wavering self-esteem, and a strong desire to fit in, body image concerns often begin in adolescence [28].

## The rise of social media

With increasing eating disorder prevalence, attention has turned to the growth of social media. In 2020, social media reached 49% of the global population [29]. Platforms including Facebook (FB), YouTube (YT), Snapchat (SC), Instagram (IG), WeChat and TikTok have created a new online world for today's youth. Recent reports reveal that 91% of UK and US adolescents use social media, with over 50% checking these at least once per hour [30]. Users can choose who to follow or message, what content to engage with or upload, what to highlight or conceal. Using filters and editing tools, individuals can alter their identities and dictate how they and their lives are perceived by others [20].

**Table 2. The complex aetiology of eating disorders.**

BIOLOGICAL FACTORS

- Genetic predisposition
- Gender: female-male ratio 10:1 for restrictive type eating disorders and 2:1 for bulimic spectrum eating disorders
- Obsessive-compulsive or autistic spectrum traits
- Susceptibility to appetite dysregulation
- Metabolic vulnerability and high BMI
- Environmental influences in the perinatal period
- Early puberty

PSYCHOSOCIAL FACTORS

- Parental eating problems or eating disorder in first-degree biological relatives
- Peer stress (e.g. bullying, weight teasing)
- Trauma
- Culture
- Internalisation of the thin / fit ideal
- Media
- Middle-to-high socioeconomic status
- Acculturation (adoption of western beauty ideal)

PSYCHOLOGICAL FACTORS

- Personality traits (rigidity, attention to detail, perfectionism, neuroticism)
- Negative emotionality
- Increased sensitivity to social ranking and threat
- Body image concerns: dissatisfaction or disturbance
- Low self-esteem
- Appearance schemas

BEHAVIOURAL FACTORS

- Extreme weight control behaviours e.g., compulsive exercise, dieting, use of laxatives and purging
- Overconcern with weight and shape
- Social isolation
- Body avoidance or checking

Adapted from [1, 13, 24].

What is posted and well-received is not coincidental–it is dynamic, shaped by broader social and cultural ideals related to beauty [31]. Online, young people are exposed to the ever-changing societal ideals of the 'desired body', with perfection as the often-unattainable end goal [9].

## Rationale for review

Body image dissatisfaction and eating disorder pathology amongst young people is rising. According to a recent UK Government report, 95% of under 18's report that they would change their appearance, and body image was one of the top three anxieties amongst Australian youths [32, 33]. An estimated 13% of young people experience an eating disorder by the age of 20, and 15–47% endorse disordered eating cognitions and behaviours [23]. Exploratory evidence indicates that social media usage may be partly to blame [34, 35].

Research has highlighted factors such as the ease of accessing harmful eating disorder-promoting content, the pervasiveness of personalised 'for you page' algorithms and the explosion of weight loss trends that inspire extreme fitness or thinness [20, 36, 37].

In parallel, recent publications have drawn attention to rising concerns of modern social media platforms and public health, and the need to understand app engagement amongst younger demographics [38].

Despite this, the association between social media, body image and eating disorders remains relatively unexplored. However, with 41.9 million neglected cases of eating disorders

in 2019, combined with unprecedented social media exposure amongst young people, this issue warrants further review from a global health perspective. Is social media a plausible risk factor for the development of body image concerns and recent rise in eating disorders? If so, is it a global or western phenomenon?

### Research aim and objectives

Our review aims to systematically map out and critically review the existing global literature on the relationship between social media, body image and eating disorders amongst young people. We provide a glossary of key terms to aid global health audiences unfamiliar with terminology related to social media, body image and eating disorders [**Table 3**].

In this review, we assess whether social media use could be a plausible and significant risk factor for the development of subclinical and clinical eating disorders on a global scale. We identify populations of young people affected, primary outcomes and any moderating or risk enhancing factors. We also explore pathways that may mediate the relationship between body image concerns and eating disorders/disordered eating behaviours within distinct social media platforms. Finally, we highlight gaps in the literature and recommend areas of focus for future research and for global health.

## Methodology

We used Arksey and O'Malley's framework and the updated PRISMA checklist for scoping reviews [39, 40] to guide our approach.

### Search strategy and study selection

We searched MEDLINE, PyscINFO and Web of Science databases in May 2021 and updated our search on 20th July 2021. We identified appropriate search terms through preliminary reading and listing relevant Medical Subject Headings. Keywords were related to four principal concepts: social media, body image, eating disorders and young people [**Table 4**]. We entered keywords manually and used the "*" symbol to capture all potential word-endings. A full search strategy for PsycINFO is in **S1 Fig**.

We exported search results from each database to the reference manager Zotero. First, we removed duplicates. Next, we screened titles and abstracts and eliminated irrelevant papers. Subsequently, we assessed full-text articles against predetermined eligibility criteria, and recorded reasons for exclusion [**Table 5**]. We identified additional studies through hand searching reference lists. Both authors independently screened titles and abstracts and reviewed full-text articles for inclusion. One reviewer carried out data extraction from all included studies, and the other reviewer independently verified that all entered information was correct. Any discrepancies were resolved by enlisting the help of additional reviewers who were not part of the study but had the relevant background in global health and nutrition.

To allow for sufficient depth of analysis and documentation of individual differences, we included studies involving young people (defined by the WHO as individuals aged 10–24 years), irrespective of gender, sexuality, ethnicity, or existing eating disorder status. If papers did not indicate the age range of participants, we included them if the reported mean age was ≤ 24. Our review was not restricted by geographic location or country income grouping, as a deliberate measure to develop a global understanding of social media use and body image or eating disorder, without making assumptions about the existence or nature of the phenomenon in countries categorised as low- or middle-income. We included studies on any social media platform (singular, multiple, or general), but not those focusing on mass media and / or internet use. We excluded studies exploring social media interventions and body image /

**Table 3. Glossary of terms associated with social media, body image and eating disorders.**

| Term | Description |
|---|---|
| Appearance comparison tendencies | The degree to which an individual tends to compare themselves to others |
| Body image | Thoughts, feelings, and perceptions related to one's body, perceived attractiveness, and self-worth |
| Bonespiration | A social media trend that idealises a very thin body through photos of people with protruding bones |
| Disordered eating | Abnormal food or eating behaviours relevant to eating disorders, such as extreme dieting, eliminating certain food groups, laxative use, binging or purging |
| Eating disorder | A group of psychiatric illnesses characterised by disturbed relationships with food, body image and exercise |
| Eating disorder pathology | Used interchangeably with disordered eating |
| Ecological Momentary Assessment | A type of study aiming to research people's thoughts and behaviours in real time by repeatedly sampling them in their natural environment |
| Facebook | A social networking app where users can create a profile, add other users as 'friends', send messages, comment or like photos, post status updates, share videos and receive notifications when other users update their profiles |
| Filter | A feature that allows you to apply pre-set edits to enhance or change a photo or video |
| Fitspiration | A social media trend aiming to inspire and motivate a healthy and fit lifestyle |
| Generation Z | Individuals born between 1997–2012 |
| Hashtag | User-generated labelling of content which aims to categorise content thematically e.g., #food, #health #fitness. Users can search hashtags to see all content related to that topic |
| Appearance focused social media | Social media platforms that use images or videos as the main mode of communication, e.g., Snapchat, Instagram, TikTok or YouTube |
| Instagram | An interactive social networking app that allows users to share videos and photos to their profiles, 'follow' and interact with other users (peers, brands, celebrities, influencers) and apply filters |
| Instagram captions | Text that appears underneath an Instagram photo or video to add context |
| Mediator | Any mechanism that may underlie the relationship between exposure and outcome |
| Moderator | Any variable that may strengthen or weaken the relationship between exposure and outcome |
| Pre-existing body image | Feelings related to one's body independent of any manipulation or association with other variables |
| Self-objectification | Occurs when an individual internalises a third-person perspective of themselves as an object to be evaluated and judged based on their appearance |
| Self-schema | The extent to which an individual cares about their appearance and how this influences their behaviour |
| Selfie | A photo taken of the self (usually with a smartphone) that can be shared on social media |
| Snapchat | An instant messaging app where conversations consist of pictures, videos and text which disappear after a set amount of time |
| Social media | User-generated platforms allowing for content creation and sharing, exchange of information, advertising and marketing, social networking, and formation of virtual communities |
| Social media influencer | Content creators with large followings and established online credibility. They may be paid by brands to advertise products or may share content aiming to influence or inspire other users (e.g. fitness, recipes, fashion, or skincare related) |
| Social networking sites | Used interchangeably with social media |
| Sociotrophy | A personality trait characterised by a need to please others and low autonomy / lack of independence |
| Thin / fit ideal internalisation | The degree to which an individual adopts and endorses the societal ideal of looking thin or fit |

(*Continued*)

**Table 3.** (Continued)

| Term | Description |
|------|-------------|
| Thinspiration | A social media trend aiming to motivate individuals to reach an extremely low bodyweight |
| TikTok | A video sharing social networking app that allows users to create and share videos with background sounds/music |
| WeChat | A Chinese mobile social media app which allows for the instant exchange of messages, voice notes, images, videos, user location sharing and the creation of a profile |
| YouTube | An online video sharing platform where users can create, watch or subscribe to content and other 'YouTubers' |

disordered eating outcomes because they were beyond the scope of our review. Papers published between January 2016 and July 2021 were eligible.

## Data extraction, charting and critical appraisal

**Data charting process.** We used a data extraction table to synthesise relevant information from each study, starting with study type, country, and World Bank Income classification [41]. We also recorded number of participants, gender (percentage female), age range (mean and standard deviation), sexual orientation and ethnicity. After testing the extraction framework on a small number of included articles, BMI and eating disorder prevalence were added as additional participant characteristics.

We extracted information on study objectives, social media platform(s), underlying theoretical framework (stated or indicated) and definitions of exposure (social media usage), outcome (body image or eating disorder / disordered eating), mediator or moderator variables.

**Critical appraisal.** Formal critical appraisal is not a requirement of scoping reviews [42]. However, we aimed to analyse the relationship between social media, body image and eating disorder pathology and the plausibility of social media as a risk factor for clinical / subclinical eating disorders to guide future global health research and policy, and thus felt that a rigorous understanding of the quality of evidence was necessary.

We used three validated critical appraisal tools to account for study design, including: (i) the Joanna Briggs Institute (JBI) checklist for analytical cross-sectional studies for cross-sectional, ecological momentary assessment, mixed methods, and longitudinal observational studies; (ii) an adapted version of the JBI tool for quasi-experimental studies for experimental

**Table 4.** Generic search terms.

| CONCEPT | SEARCH TERMS |
|---------|--------------|
| Social Media | "social media" or "social networking site*" or Instagram or TikTok or Snapchat or Facebook or Twitter or Pinterest or YouTube |
| Body Image | "body image" or "body perception" or "body surveillance" or "body shame" or "body comparison" or "body checking" or "body image avoidance" or "body image disturbance" or "body dissatisfaction" or "body negativity" or "negative affect" or "body positivity" or "body acceptance" or "body appreciation" or "positive affect" or "body satisfaction" |
| Eating disorders | "eating disorder*" or "disordered eating" or "anorexia" or "bulimia" or "binge eating disorder*" or OSFED or "orthorexia" or "body dysmorphia" or "atypical anorexia" or "compulsive exercise" or "extreme dieting" or "clean eating" or binging or fasting or overeating or undereating or purging or fitspiration or thinspiration or "thin ideal" or "pro-eating disorder" or pro-anorexia or pro-bulimia |
| Young people | "youth*" or "adolescent*" or "young adult*" or "teen*" or "young people" or "young person" |

**Table 5. Inclusion and exclusion criteria.**

| INCLUSION | EXCLUSION |
|---|---|
| **All studies** | |
| • English language<br>• Peer reviewed papers<br>• Published between January 2016 and July 2021<br>• All study types, including previous systematic reviews and meta-analyses<br>• All social media platforms (singular, multiple, or general)<br>• All geographical locations<br>• Any study setting: community based, clinical, online<br>• Participants aged 10–24 or with a combined mean age of ≤ 24<br>• All subgroups of young people (gender, sexuality, ethnicity, eating disorder status)<br>• Exploration of social media use and body image / disordered eating outcomes, and relevant mediating or moderating factors | • Theses, non-academic grey literature, books, and book chapters<br>• Studies with no full text available<br>• Focus on mass media or general internet use rather than social media<br>• Focus on social media related interventions for eating disorders |
| **Additional criteria for quantitative studies** | |
| • Appropriate measure of social media use and valid tools to confirm body image or disordered eating outcomes | N/A |
| **Additional criteria for qualitative studies** | |
| • Exploration of themes related to social media use and body image/ disordered eating | N/A |

and mixed methods experimental studies; and (iii) the Critical Appraisal Skills Checklist (CASP) for qualitative studies.

We gave studies one point for every checklist item fully met, and half a point when the item was partially met, and then calculated the proportion of checklist items met by each study. We categorised study quality as High (≥ 75% checklist covered), Moderate (50%-74% checklist covered), and Low (<50% checklist covered).

## Theoretical framework

Our review and appraisal of the evidence was informed by Rodgers' 2016 integrated theoretical framework [43] on the influence of internet use on body image concerns and disordered eating pathology. Interactions with others as well as individual online behaviours are important pathways linking internet use to body image and eating pathology, and a hypothesised feedback loop between the two reinforces the addictive nature of social media and sustained motivations for use despite potentially adverse outcomes. Whilst previous frameworks have focused on singular theoretical perspectives that explain this relationship, Rodgers incorporates five theories to provide mechanistic insight, including (i) sociocultural theory, (ii) self-objectification and feminist theory, (iii) impression management involving self-discrepancy and true-self theories, (iv) social identity theory, and (v) gratification theory.

i. *Sociocultural Theory* postulates that social agents (media, peers, and family) convey a strong need to conform to societal body ideals [44]. Comparisons to similar others online and the internalisation of the thin ideal may lead to binge eating disorder and compensatory disordered eating behaviours [45].

ii. *Self-objectification and Feminist Theory* hypothesises that in an appearance obsessed society, women are acculturated to internalise a third person perspective of themselves, leading to habitual body monitoring, dissatisfaction, and subsequent disordered eating behaviours [46]. Social media offers opportunity for posting self-images, which are then subject to scrutiny and feedback.

iii. *Impression Management (self-discrepancy and true self theories)* emphasises that individuals seek to control their identity in line with societal ideals [47]. On social media, individuals can curate a perfect online persona—their 'ideal self'. The discrepancy between the ideal self and less perfect offline self is thought to induce body image concerns. Compensatory disordered eating behaviours may be employed to move closer to the online ideal.

iv. *Social Identity Theory* posits that individual identity is drawn from group membership with like-minded individuals. On social media, opportunities to be part of 'in groups' are endless, irrespective of physical proximity. Examples include weight loss groups, healthy eating and fitness communities, pro-eating disorder pages, and eating disorder recovery groups. Shared values of online communities may influence body image and eating behaviour via perceived consensus and 'in group' norms.

v. *Gratification's Theory* restores the agency of individuals in seeking out certain social media content in accordance with inherent personality traits, needs, and interests. Individual characteristics, heterogenous motivations for social media use and specific online behaviours may heighten vulnerability to subsequent body image concerns and disordered eating behaviours.

We used Rogers' framework in our scoping review in two ways. First, we assessed whether each included article addressed any of the framework's specific pathways and noted any underlying theoretical assumptions that were outside of Rogers' model but could potentially extend it. Second, we evaluated included studies to understand how well the current scope of literature matches Rogers' proposed view, and how we could extend and update the integrated model in light of emerging evidence on social media use, body image and disordered eating published since 2016.

## Ethics

The review was based on previously published studies and therefore no ethical approval or participant consent was required.

## Results

This section details the findings of our review, beginning with an overview of search results and study characteristics. We then summarise the main exposures, outcomes, mediators and moderators identified in the literature. We present our synthesis of the literature in a framework describing a self-perpetuating cycle of risk.

## Study selection

We identified 273 articles from database searches, and after de-duplication, screening abstracts, reading full-texts, and hand-searching reference lists, we included 50 studies (45 quantitative and 5 qualitative) in our review (Fig 1) (See **S1 Table** for individual summaries of the 50 studies).

## Study characteristics

**Study design.** We reviewed 45 quantitative (30 cross-sectional, 6 experimental, 5 mixed methods, 2 ecological momentary assessment, and 2 longitudinal observational studies) and 5 qualitative studies. Qualitative studies collected data through focus group discussions (n = 4) and interviews (n = 1).

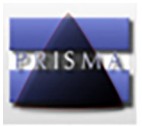

## PRISMA 2009 Flow Diagram

**Identification**

Records identified through database searching (n = 273)
Medline (n = 79)
PyscInfo (n = 75)
Web of Science (n = 119)

Additional records identified through other sources (n = 3)

Records after duplicates removed (n = 194)

**Screening**

Records screened (n = 194)

Records excluded (n = 50)

**Eligibility**

Full-text articles assessed for eligibility (n = 144)

Full-text articles excluded, with reasons (n = 94)

- No validated scale for assessing exposure & outcome (n = 15)
- Content analysis (n=9)
- Out of age range (n=13)
- Thesis (n=3)
- No full text available (n=5)
- Published prior to 1/1/16 (n=17)
- Not a study (n=7)
- Not in English (n=3)
- Media or general internet (n=6)
- SM intervention (n=8)
- Lack of relevance to research question (n=8)

**Included**

Studies included in synthesis
Quantitative (n=45)
Qualitative (n =5)

**Fig 1. PRISMA flow diagram.**

**Study location.**   Ninety percent of studies (n = 45) were conducted in high-income countries. The largest number of studies were conducted in Australia (n = 12) and the United States (n = 11). Canada, Italy, Singapore, and United Kingdom were included in three studies each, Spain and Ireland in two each, and Belgium, France, Germany and Sweden in one each. Studies conducted in Sri Lanka (n = 1), China (n = 2), Malaysia (n = 1) and Thailand (n = 1) comprised the evidence base on upper-middle-income and lower middle-income countries, and none included populations in low-income countries.

**Participants.**   48% of studies (n = 24) included only female participants, 2% of studies (n = 1) included male participants, and 50% (n = 25) included both genders. One study reported transgender participants, although this subcategory constituted only 9.3% of the study's total sample [48].

Where ethnicity was reported, >75% of studies (n = 18) reported majority White participants, 17.4% (n = 4) reported majority Chinese participants, and <5% reported majority Hispanic participants.

Only 17 studies described participants' BMI. Most participants (82%) were of a healthy BMI (18.5 to 25), with smaller proportions who met criteria for overweight (12%) and underweight (6%) BMI categories. Four studies (8%) identified participants with an existing eating disorder.

Most studies recruited participants from university (n = 20) and secondary schools (n = 19), with fewer studies using online (n = 6), community (n = 4) and clinical (n = 1) settings to recruit young people.

**Underlying theoretical frameworks.**   Most studies borrowed ideas from multiple theories. 78% of papers referred to Sociocultural Theory, with an emphasis on social comparison. 26% referred to Gratification's Theory, 24% reported Self-objectification and Feminist Theory, 24% highlighted Impression Management Theory, and 12% of studies mentioned Social Identity Theory. We identified seven theories outside of Rodgers' framework [S2 Table].

**Social media platforms assessed.**   Twelve studies assessed general social media usage, 18 indicated multiple platforms, and nine focused on appearance orientated sites only (Facebook, Instagram, Snapchat, YouTube). Where specified, most studies investigated Instagram (n = 15), followed by Facebook (n = 5) and WeChat (n = 1).

**Motivation for social media usage.**   Across studies, reasons for social media usage included: identity management, fitting in with friends, posting content for peer feedback, and seeking out weight loss, fitness, or pro-eating disorder material.

**Quality appraisal.**   Many quantitative studies (n = 28) were of moderate quality. Generally, unfulfilled criteria included failure to address confounding variables, and lack of control group in experimental conditions. Most qualitative studies (n = 4) comprised high-quality evidence (>75% checklist met), although they did not always consider ethical issues or the positionality of the researcher [S3–S5 Tables].

## Relationships between social media, body image and eating disorders

The overarching relationship between social media use, body image and eating disorders operates through a range of mechanisms. Typically, social media usage led to body image concerns, eating disorder or disordered eating outcomes, and poor mental health via the mediating pathways of social comparison, thin / fit ideal internalisation, and self-objectification.

## Exposures

**Specific social media usage.**   Approximately 58% of studies (n = 29) investigated specific types of social media exposure, including time, frequency, use of appearance-focused platforms, and investment in appearance related activity [Table A in S1 Data].

**Time.** Seven studies investigated the relationship between time spent on social media and body image or eating disorder-related outcomes. Time was significantly associated with these variables in two studies, although both papers failed to acknowledge other social media activities and possible mediators [49, 50]. Three cross-sectional studies discovered that time spent on social media was associated with body image dissatisfaction via the mediating pathways of social comparison and thin ideal internalisation [51–53], indicating that the relationship between exposure and outcome is more nuanced than the mere number of hours spent online.

**Frequency.** High frequency of social media usage and body image dissatisfaction was supported by two studies [54, 55].

**Appearance focused social media platforms.** Three cross sectional studies indicated that appearance focused platforms, namely Instagram and Snapchat, are significantly associated with body image concerns, eating disorder pathology, anxiety and depressive symptoms [51, 56, 57].

**Investment in appearance related activities.** 17 studies identified that investment in appearance related activities ('selfie' avoidance, manipulation and posting edited photos, and significantly investing in 'likes' and 'comments') may be noteworthy exposures. These activities were consistently associated with body image dissatisfaction and risk of eating disorder pathology across a range of cross-sectional, experimental, and qualitative study designs (n = 14).

There were anomalies to this trend (n = 3). First, two studies found that posting 'selfies' on Instagram led to higher body esteem, rather than body image dissatisfaction [58, 59]. However, included participants may have had significantly higher pre-existing body esteem, rather than this emerging as a consequence of posting.

**Social media trends.** Two prominent hashtags featured in the literature, drawing on the idea of social media as a source of inspiration or aspiration towards greater fitness (#Fitspiration) and thinness (#Thinspiration) ([Table B in S1 Data]).

*#Fitspiration*. Eight studies investigated the impact of the fitspiration trend on body image dissatisfaction and eating disorder pathology with mixed results: 50% supported the relationship, 25% partly supported it, and 25% refuted it. Three moderate- to high- quality experimental studies demonstrated that exposure to fitspiration imagery relative to control images resulted in body image dissatisfaction and negative mood for participants, pointing towards a causal relationship [60–62]. Qualitative insight highlighted that for some, fitspiration inspired healthy eating and exercise. Others felt extreme pressure to 'eat clean' or exercise to excess, with subsequent bingeing and disordered eating outcomes [63, 64]. A mixed methods study of fitspiration followers on Instagram found that 17.7% were at risk of developing an eating disorder, 17.4% demonstrated high levels of psychological distress, and 10.3% displayed addictive levels of physical exercise [65].

*#Thinspiration and pro-eating disorder content.* Three studies explored the relationship of the #thinspiration trend with body image and eating disorders. A mixed methods study concluded that the hashtag glorified "emaciated people" and "bone-thin girls", promoting starvation as a lifestyle choice instead of a symptom of mental illness. Posts provided individuals with tips on how to lose weight and hide an eating disorder [48]. A cross-sectional study found that 96% of included participants followed the thin-ideal on social media, of whom 86% met the criteria for a clinical/subclinical eating disorder, and 71% and 65% reported symptoms of depression and anxiety, respectively [66]. Whilst these statistics are alarming, the study relied on self-reported symptoms and used novel eating disorder diagnostic tools that had not been extensively validated.

## Outcomes

**Eating disorder pathology.** 20 studies explored eating pathology as an outcome of social media usage [**Table C in S1 Data**].

**Clinical/subclinical eating disorders.** Five cross-sectional studies yielded statistically significant associations between social media usage and various clinical eating disorders. These ranged from night eating syndrome [67], to binge eating disorder [68] and bulimia nervosa [69].

One cross-sectional and one qualitative study indicated orthorexia nervosa symptomatology amongst participants, ranging from obsessions with 'clean eating' to avoidance of 'demonised' foods and compulsive exercise behaviours [64, 70]. A study of 713 participants confirmed orthorexia nervosa prevalence of 49% [70], far greater than the estimated <1% in the general UK population. However, participants were recruited from 'fitness' Instagram pages, and thus unlikely to be representative of all social media users or the general population.

**Disordered eating behaviours.** More commonly, 11 studies found statistically significant associations between social media usage and disordered eating behaviours, including bingeing, purging, use of laxatives and extreme dieting. One cross-sectional study found that 51.7% of adolescent girls and 45% of boys engaged in meal skipping and excessive exercise [57]. Although the sample size was large (n = 993), behaviours were self-reported and study quality was low.

**Eating disorder maintenance or recovery.** Two studies explored the effect of social media on eating disorder maintenance or recovery. A mixed-methods study found that only 3% of 499 participants with clinical/subclinical eating disorders used social media to aid recovery or as a form of treatment. The remaining 97% indicated that it hampered recovery, one stating that "when I get really hungry, I go into these sites to get motivation to not eat for a bit longer" [48].

**Body image concerns.** 33 studies demonstrated significant associations between social media usage and body image dissatisfaction, including body shame, low self-esteem and body related anxiety [**Table D in S1 Data**]. Of these, five hypothesised that body image dissatisfaction preceded subsequent eating disorder pathology [50, 71].

**Mental health.** Although not the primary focus of the research, nine studies revealed significant associations between social media usage, body image concerns or disordered eating pathology, and poor mental health [**Table E in S1 Data**]. Outcomes included low mood (n = 4), anxiety and depressive symptoms (n = 5).

## Mediators

We identified three key mediators [**Table F in S1 Data**].

**Thin / fit ideal internalisation.** 12 studies investigated thin / fit internalisation as a mediator between social media usage and body image or disordered eating outcomes. Eleven (92%) indicated that it is a plausible mediator, across cross sectional (n = 7), experimental (n = 1) and qualitative (n = 3) study designs. In a qualitative study, female participants (n = 27) in focus group discussions reported feeling pressure to adhere to an ever-changing ideal [72]. In qualitative interviews, a sample of Swedish adolescents reported feelings of alienation following failure to adhere to the 'toned but not too muscular' ideal, whilst focus groups with Irish adolescents revealed participants' feelings of self-blame and disgust [64, 73].

**Appearance comparisons.** 21 studies explored the mediator of appearance comparisons on social media, with 19 (90%) reporting a significant relationship. Comparisons tended to be 'upward' and yielded feelings of inadequacy and self-loathing [53, 63, 68]. In contrast, an observational longitudinal study revealed that comparisons on Facebook did not predict body image dissatisfaction six months later [74]. However, Facebook use was marked as 'outdated' amongst younger participants (mean age in this study was 14.7 years) which could account for the non-significant finding.

**Self-objectification.**    Six studies recognised self-objectification as a significant mediator. Generally, participants reported self-criticism, picking out flaws in photos and purposively posting photos accentuating certain body parts [48, 51, 73].

## Moderators

The relationship between social media and body image / disordered eating was inflected by several moderating factors, broadly categorised as biological [**Table G in S1 Data**], cognitive [**Table H in S1 Data**] and socio-environmental [**Table I in S1 Data**] characteristics.

### Biological

**Gender.**    18 studies investigated gender as a moderator, 14 of which found significant differences between males and females. Generally, girls invested heavily in photos of themselves, endorsed the thin / fit ideal, made more comparisons with others, and engaged in higher levels of disordered eating pathology, specifically dieting and emotional eating [50, 57, 75]. In contrast, boys endorsed a more muscular ideal, with goals of functionality and fitness, rather than weight loss [76, 77].

Qualitative focus groups implied that social media and body image is perceived as a gendered subject, with boys feeling reluctant to admit adverse effects due to stigma or fear of emasculation [63]. As most studies implemented self-report tools, and over half comprised mixed gender participants, this may have distorted findings. Remaining studies (n = 4) found no difference between males and females. Discrepancy may be due to variance in the assessment tools utilised for body image or disordered eating outcomes and differential sex ratios and mean age of participants.

**BMI.**    Five studies investigated BMI as a moderator. Three indicated high BMI as strengthening the relationship between social media, body image dissatisfaction and eating disorder pathology. The two anomalies to this trend both included participants with abnormally low average BMIs [71, 78]. All studies relied upon self-reported BMI, thus increasing risk of measurement error and bias.

### Cognitive

**Pre-existing body image concerns.**    Four studies indicated that inherent body image concerns (shame and low self-esteem) predicted certain social media behaviours and heightened susceptibility of eating disorder pathology. In contrast, body appreciation appeared to buffer against this effect [71], however, self-compassion was not a moderator [79].

**Risk of eating disorder.**    Five studies highlighted that high-risk individuals with elevated eating disorder scores or a pre-existing eating disorder are more inclined to seek out damaging content on social media (such as thinspiration or weight loss). They may be more susceptible to mediating factors (internalisation and comparison) and are thus at heightened risk of further clinical / subclinical eating disorders [78, 80, 81].

### Socio-environmental factors

**Social media literacy.**    Three studies indicated low social media literacy amongst participants, including difficulty switching off from damaging posts and an inability to recognise edited versus unedited posts [63, 73, 82]. In contrast, focus groups with adolescent girls in the US showed that having studied social media in school, girls were critical of its artificiality, reported that they could appreciate others' beauty without jealousy, were self-accepting and

did not feel the need to seek compliments online. Some engaged in comparisons, although this did not appear to lead to body image dissatisfaction and eating disorder pathology [83].

## A self-perpetuating cycle of risk?

Our synthesis indicates that certain social media exposures and individual risk factors can strengthen this relationship, whilst numerous moderators may weaken, or even disrupt it. This led us to extend Rodgers' framework [43] in light of new research, referred to here as a 'self-perpetuating cycle of risk' [**Fig 2**].

Our findings indicate that specific features of social media usage (appearance focused platforms, investment in photos, and engagement with fitspiration and thinspiration trends) lead to body image concerns, disordered eating pathology and mental health outcomes. This relationship is shaped by the mediating pathways of thin / fit ideal internalisation, appearance comparisons, and self-objectification, which have been supported by additional meta-analyses [3, 84, 85].

However, due to the cross-sectional nature of most studies, it is impossible to identify the direction of causality: for example, do body image dissatisfaction and disordered eating occur because of social media usage, or do these pre-exist, encourage engagement in certain online activities, and result in unfavourable clinically significant outcomes?

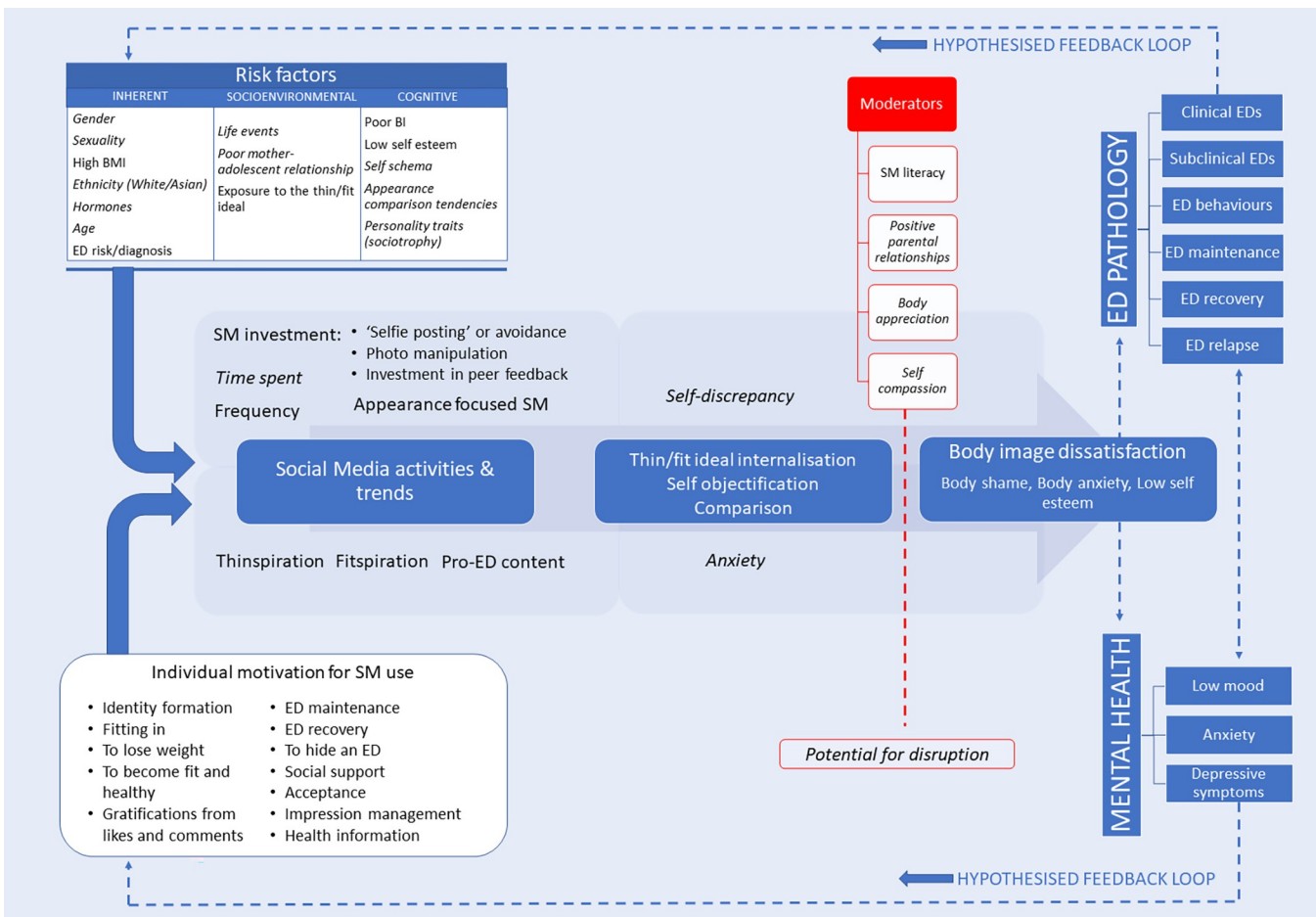

**Fig 2. A self-perpetuating cycle of risk to show the relationship between social media usage, body image and eating disorder pathology.**

Our revised framework recognises both possibilities. It is plausible that specific individual risk factors (particularly high BMI, poor body image and existing eating disorders) combined with differential motivations for social media use (identity formation, gratifications from peer feedback) encourage certain behaviours when users engage with social media (photo manipulation, searching for thinspiration and fitspiration content), strengthen the effects of mediators, and increase risk of poor body image, disordered eating, and mental health outcomes. This shifts away from the fatalistic notion that social media causes poor body image and eating disorders in all users. Instead, it suggests that certain individuals are simply more vulnerable to its deleterious effect.

This relationship is not linear. Results highlight that social media is highly addictive, and individuals use it despite negative outcomes [59]. In fact, to 'fix' their poor body image, users may be even more inclined to do so (e.g., manipulate photos to obtain more likes)—indicated by a hypothesised feedback loop. It is this which may trigger the self-perpetuating cycle of risk. However, this cycle can be broken. Several moderators, or buffers, that have the potential to disrupt it. Many studies showed that whilst individuals still internalised the ideal or compared themselves to others, high social media literacy and body appreciation prevented this from resulting in body image dissatisfaction, disordered pathology and poor mental health [71, 83].

## Discussion

Our scoping review of 50 studies conducted in 17 countries found that social media usage is a plausible risk factor for the development of clinical / subclinical eating disorders across a range of country income groupings. We extended current thinking to describe how social media provides a platform of perfectionism, often embeds unhealthy ideals of disordered eating and fitness and can hamper recovery from eating disorders. We also identified mediators, moderators and important risk factors that shape this relationship and offer opportunities to intervene.

Taken as a whole, the literature underlines a complex, yet meaningful relationship between social media usage, body image concerns, and disordered eating pathology. While our review points to potentially large scale implications among the approximately 3.9 billion social media users worldwide, it is important to note that not every user has poor body image or an eating disorder [29]. This begs the question, what makes certain individuals more susceptible?

### Significance for global health

Disturbingly high prevalence of body image dissatisfaction, disordered eating pathology and comorbid mental health outcomes were reported amongst young social media users in this review [57, 66, 70]. Given the sheer scale of social media reach (approximately 60% of the world's young people), a large proportion of young people could be exposed to the self-perpetuating cycle of risk [29]. Significantly, findings were reflected amongst Asian samples, mirroring concerns regarding the explosion of eating disorders in Asia [86, 87] which has a population of 4.6 billion. Social media use as a risk factor for eating disorder pathology clearly warrants attention outside of high-income western countries [17].

Currently, up to 80% of eating disorder sufferers remain out of formal healthcare systems, with many presenting at late stages [66]. Denial, stigma, and fears that one's disorder isn't serious enough already hampers treatment seeking [14]. Today, young people are often immersed in a digital world where desires to change one's body, excessive exercise and preoccupation with food appears normal [88]. How can young people identify that they have a problem, when their behaviours seem to be nothing out of the ordinary?

Eating disorders, clinical or subclinical, are serious psychiatric disorders with a range of comorbid health outcomes [28, 89]. Unfortunately, they are often misunderstood, omitted from nationally representative health surveys, and viewed as less important than other mental health disorders [12]. Reports of 41.9 million neglected eating disorder cases in 2019, combined with a surge in cases recorded by health systems and charities alike calls for a serious reconceptualisation of the disorder [36, 90].

This issue also sits within the wider arena of adolescent mental health. The WHO Global Strategy for Women's, Children and Adolescent Health coupled with The Mental Health Action Plan 2013–2020 demonstrate that investment in young people and their mental health yields invaluable gains for society [91, 92]. Although eating disorders were recognised by The National Institute of Mental Health as a priority area for adolescents in 2007 with prevalence equalling that of bipolar and substance use disorders, they remain absent from these seminal reports and constitute a fraction of global mental health research [12, 93, 94].

Since 2007, the rise of social media has brought new challenges. Despite its dominance in the lives of Generation Z, we are only just beginning to learn about its impact [95, 96]. Of significance, The UK Royal Society for Public Health (RSPH) published a damning report on social media and mental health, whilst social media, body image and eating disorders have been recognised as emerging policy areas for the UK, Australian and US government alike [97–100]. In 2020, The UK launched their Online Harms White Paper to promote a regulatory framework of online safety and stimulate innovative social media intervention to protect young users. Eating disorder risk, however, remains absent from many of these youth centred goals in the UK [101].

Despite this rising concern, regulation of social media remains weak, with significant gaps between 'safety policies' and the real-life experiences of users [80]. Age, anonymity, and the pervasiveness of algorithms play a key role in this. Although required that users must be at least 13 years of age, most popular social media platforms have no robust means of age authentication. Recent figures reveal that up to 42% of children under the minimum age have a social media profile [102]. Users of all ages can join under any email address and disguise their identity through use of aliases. Once online, access to content is generally unrestricted, whilst algorithms suggest personalised content based on prior user engagement. In April 2021 Instagram sparked headlines after 'appetite suppressants', 'fasting' and 'weight loss' was recommended to certain users based on their previous searches [103]. Although rectified, content that glorifies eating disorders remains highly accessible, with little promise of intervention. More critically, internal research by Facebook conducted in 2019 leaked to the media in September 2021 showed not only that 40% of Instagram users who reported that they felt unattractive said that dissatisfaction began while using the platform [104], but also suggests that Facebook knew about the app's potential to harm teenage girls' mental health [105].

Further to this, evidence suggests that social media literacy amongst the young is low, and clinical recognition of eating disorder symptomatology and online risk factors poor [106]. In parallel, social media has been described as more addictive than alcohol and cigarettes, body image dissatisfaction and eating disorder pathology are on the rise, and the mental health of today's youth is declining [98]. In light of COVID-19 lockdowns and further shifts towards an online world, this issue demands greater global attention [55].

## Recommendations and future research

As the topic is in its infancy, our recommendations for intervention and future research are generally exploratory.

## Community level

Primarily, opportunity lies in raising awareness of social media and its possible connection to body image dissatisfaction and eating disorder symptomatology. In the UK, only 23% of young people learn about body image at school, although 78% believe that this would be useful [32]. Through investments in social media literacy [83], young people could learn to appreciate body diversity, navigate social media through a critical lens, and challenge the artificiality of the societal ideal of beauty. Open discussions between students, teachers, and parents could reduce the stigma associated with eating disorders and help young people to identify body image concerns and eating problems before they manifest as serious disorders. Preliminary research investigating social media literacy has shown promising outcomes [107].

## Societal and health systems level

At a societal level, recognition of the issue within government and health systems is paramount. To optimise positive social media use, communication between these players and social media companies should be enabled. From this, policies to enhance age verification, minimise access to pro-eating disorder content and increase the health and safety of users could be developed [106]. Recently, the UK Government Equalities Office approached social media influencers and advertising companies to devise strategies to enhance body diversity online [97]. Results are yet to materialise, although there is evidence of an emerging online 'body positivity' movement [95, 108].

Within a clinical setting, appreciating eating disorders as serious mental health problems, screening patients presenting relevant symptomatology regardless of body size, and integrating social media as an additional factor within treatment plans could be advantageous [109]. At the global level, we recommend building health system capacity in low and middle-income countries in preparation for the potential future burden of eating disorders as a progressive strategy. However, based on funding, budget cuts, and the priority that other health issues currently take, the likelihood of this being implemented is unclear [110].

## Study design

As discussed, the scope of evidence in this review is limited by the cross-sectional design adopted by most studies, the homogeneity of included participants, and limited geographic scope. Moving forward, researchers should consider conducting longitudinal studies with representative samples and cross-cultural comparisons covering all regions. This would provide greater clarity regarding the true directionality of the relationship, and grant insight into the impact of social media on young people across the life course. Likewise, ongoing qualitative studies with young people would aid understanding and provide rich data on a topic that is unique to this age group.

While our review describes certain risk factors, mediators and moderators in the extended framework, the nature of their role was not conclusive (including gender and personality traits). Future research would benefit from focusing on these to see whether they are as meaningful as current evidence suggests. The increase in eating disorder incidence among males and gender differences described by a small number of studies suggest that this particular moderator requires further investigation [111].

Next, previous research has highlighted that certain subcategories, including those with high BMI, athletes, and young people who identify as lesbian, gay, bisexual and transgender (LGBT) are more susceptible to body image dissatisfaction and eating disorders [112]. Study populations in this review tended to be White school or university students and did not reflect

the intersectionality nor diversity of national populations. Thus, it would be beneficial to assess the relative impact of social media on eating disorders across these high-risk groups.

Finally, prior evidence suggests that media penetration and adoption of 'western ideals' increases the risk of eating disorder pathology [113]. Despite global scope, this review did not retrieve any studies from low-income countries. Therefore, this assumption requires further testing in these contexts. Future research should explore the relationship in low income, non-western cultures, which may be important as social media use rises globally.

## Limitations of this review

First, due the cross-sectional nature of most included studies and generally 'moderate' quality, causation between social media usage and outcomes cannot be presumed. Likewise, with variance in tools used to assess mediators and outcomes, most of which were self-reported, measurement error and bias are possible.

Secondly, despite the reviews' inclusive geographic scope, studies were concentrated within middle-high income countries in Europe, Asia, and Australasia. Although few cross-cultural differences emerged, scarcity of evidence from low-income countries means that generalisability is indicative rather than conclusive. Likewise, included samples were relatively homogenous. Most participants were of White ethnicity, average BMI, and female. Paucity of diverse evidence limits meaningful insight into any subcategories of particular risk.

Next, although the social media platforms covered in the literature were extensive, the digital landscape evolves rapidly. For example, TikTok (the most downloaded mobile application in 2020) is the focus of significant media concerns regarding body image yet is only mentioned by one study [55]. It may be that findings still fail to capture the most recent trends.

## Conclusion

In the 21st century, social media use amongst a developmentally susceptible age category is unprecedented and largely unregulated. 'Likes' and comments can validate identity, the societal ideal of beauty appears ubiquitous, and most people (albeit enhanced and filtered) appear to be perfect. In pursuit of acceptance, popularity and validation, the common option is to follow suit–to manage one's own online identity to meet the ideal marked by others, to manipulate and scrutinise 'selfies', and once posted, angst over the numbers of likes or comments received. However, despite one's best efforts, this online change is rarely good enough. Through the lens of social media, someone else can always look better, skinnier, or prettier [73]. Likewise, pro-eating disorder content is rife, and the 'healthy' #fitspiration trend may be fuelling new waves of disturbed eating and exercise pathology. The outcome is a population of young people at risk of corroded body image, gaping discrepancies between their actual and 'polished' online selves, and an increased likelihood of engaging in compensatory disordered eating behaviours, as our review has shown.

In parallel, cases of eating disorders are escalating, with prevalence far exceeding what was previously thought. Although it is not possible to isolate one single cause, the plausible link between social media, body image dissatisfaction and eating disorders is alarming. Based on the scale of social media usage, this could impact the wellbeing of a significant proportion of the world's young- particularly those who are already vulnerable. Where the body of evidence is so recent, and social media ever evolving, the ramifications of this are not yet fully clear.

However, with significant strides being made in the realm of global adolescent mental health, intervention is clearly possible. Through recognition, funding, research, and prioritisation, there is hope that this issue will receive more attention, and concern will translate into tangible action. Our goal should be to have a generation of young people who are body

positive, who use social media in a progressive way, who eat food because it is a basic human need, and who do not measure self-worth by the circumference of their thighs. We must also aim for widespread education and early identification of at-risk individuals, so that eating disorder symptomatology can be challenged long before it presents at formal healthcare settings.

The burden of body image dissatisfaction and eating disorders as a global health issue has been ignored for too long. The rise of social media has reinforced the need to turn attention to the 'Cinderella disease' of mental health, which this scoping review has demonstrated as worthy of a place on the global public health agenda.

## Supporting information

**S1 Fig. Full search strategy carried out in PyschINFO.**
(TIF)

**S1 Table. Summaries of individual studies.**
(PDF)

**S2 Table. Additional theories outside of Rodgers' framework.**
(PDF)

**S3 Table. The Joanna Briggs Institute (JBI) checklist for analytical cross-sectional studies.**
(PDF)

**S4 Table. Adapted version of the Joanna Briggs Institute tool for quasi-experimental studies.**
(PDF)

**S5 Table. The Critical Appraisal Skills Checklist (CASP) utilised for qualitative studies.**
(PDF)

**S1 Data. Tables A to I.** Data categorisation tables for the synthesis of results.
(PDF)

## Acknowledgments

We are grateful to Delan Devakumar for comments on an earlier draft of the manuscript.

## Author Contributions

**Conceptualization:** Alexandra Dane, Komal Bhatia.

**Data curation:** Alexandra Dane, Komal Bhatia.

**Formal analysis:** Alexandra Dane.

**Methodology:** Alexandra Dane, Komal Bhatia.

**Supervision:** Komal Bhatia.

**Validation:** Komal Bhatia.

**Visualization:** Alexandra Dane.

**Writing – original draft:** Alexandra Dane.

**Writing – review & editing:** Alexandra Dane, Komal Bhatia.

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
