## [Decision Letter · Decision Letter 0]

2 Jan 2023

PGPH-D-22-01315

The Social Media Diet: A scoping review to investigate the association between social media, body image and eating disorders amongst young people

Dear Dr. Bhatia,

Thank you for submitting your manuscript to PLOS Global Public Health. After careful consideration, we feel that it has merit but does not fully meet PLOS Global Public Health’s publication criteria as it currently stands. Therefore, we invite you to submit a revised version of the manuscript that addresses the points raised by both the reviewers.

We look forward to receiving your revised manuscript.

Kind regards,

Rakesh Singh

Academic Editor

Journal Requirements:

2. Please provide separate figure files in .tif or .eps format only and remove any figures embedded in your manuscript file. Please also ensure that all files are under our size limit of 10MB.

Reviewers' comments:

Reviewer #1: Overall, this is a well-written manuscript with rigorously methodology.

Specific comments are follows:

#Figure 1: PRISMA flow diagram

Please explicitly mention the "other sources" from where you have identified three additional records (wherever applicable).

#Recommendations and future research

page 29, line 597: Societal and health system level: The recommendations for health system level seems to be vague and focused mainly on clinical, policy, and global level. Please provide the recommendation from lowest tier to highest tier of health care system. It is not necessary to make different sections but the recommendations should be clear and detailed.

#Others

page 5, line 87, Table 2: check spelling of psychosocial

Reviewer #2: The subject is fascinating and significant from the standpoint of public health. But improvements are required to make it better. The suggestions are as follows:

• Please include the terminologies/keywords used for the search

• Line 47: There are more than six categories in table 1. What are the other categories? This is not clear. Please clarify

• Table 2: The information presented here looks more suitable for a figure than a table

• Line 134: Why is named as a Panel. Can’t this be a table?

• It will be easy for the reader to understand if the summary of the theoretical framework can be depicted in the form a diagram within the article.

• It may be better if the PRISMA checklist was included in the text itself rather than in the supplementary files.

• The results part quite lengthy and has a lot of sections and sub-sections. I will be easier to follow if a few lines are written at the beginning of the results describing how you have divided the findings.

---

## [Editor Report · Decision Letter 1]

2 Feb 2023

The Social Media Diet: A scoping review to investigate the association between social media, body image and eating disorders amongst young people

PGPH-D-22-01315R1

Dear Dr Bhatia,

We are pleased to inform you that your manuscript 'The Social Media Diet: A scoping review to investigate the association between social media, body image and eating disorders amongst young people' has been provisionally accepted for publication in PLOS Global Public Health.

Best regards,

Rakesh Singh

Academic Editor
